# Anti-AMPA Receptor Autoantibodies Reduce Excitatory Currents in Rat Hippocampal Neurons

**DOI:** 10.3390/ph16010077

**Published:** 2023-01-04

**Authors:** Charlotte Day, John-Paul Silva, Rebecca Munro, Terry S. Baker, Christian Wolff, Angela Bithell, Gary J. Stephens

**Affiliations:** 1School of Pharmacy, University of Reading, Whiteknights, Reading RG6 6AJ, UK; 2UCB Pharma, 208 Bath Road, Slough SL1 3WE, UK; 3UCB Pharma, Chemin du Foriest, B-1420 Braine l’Alleud, Belgium

**Keywords:** AMPA receptor autoantibodies, excitatory currents, hippocampal neuron

## Abstract

The GluR3 subunit of α-amino-3-hydroxy-5-methyl-4-isoxazolepropionic acid receptors (AMPARs) has been identified as a target for autoantibodies (Aabs) in autoimmune encephalopathy and other diseases. Recent studies have proposed mechanisms by which these Aabs act, but their exact role in neuronal excitability is yet to be established. Patient Aabs have been shown to bind to specific regions within the GluR3 subunit. GLUR3B peptides were designed based on described (ELISA) immunogenic epitopes for Aabs and an immunisation strategy was used to generate novel anti-AMPAR Aabs. Target-specific binding and specificity of affinity-purified anti-AMPAR Aabs was confirmed using enzyme-linked immunosorbent assay, immunocytochemistry and Western blot. Functional anti-AMPAR Aab effects were determined on excitatory postsynaptic currents (EPSCs) from primary hippocampal neurons using whole-cell patch-clamp electrophysiology. Acute (10 or 30 min) or longer-term (24 h) application of anti-AMPAR Aabs caused a significant reduction in the mean frequency of spontaneous and miniature EPSCs in hippocampal neurons. Our data demonstrate that anti-AMPAR Aabs targeting peptides linked to auto-immune diseases mediate inhibitory effects on neuronal excitability at the synaptic level, such effects may lead to disruption of the excitatory/inhibitory balance at a network level.

## 1. Introduction

In autoimmune conditions, a breakdown in self-tolerance and a persistent immune response against self-proteins is observed, resulting in the production of Aabs [1,2]. Currently, more than 2.5% of the population is estimated to be affected by an Aab-driven autoimmune disease, although with increasing knowledge, this percentage is rapidly rising [3]. In particular, Aabs against central nervous system (CNS) targets including neurotransmitter receptors such as AMPARs have been implicated in pathological auto-immune conditions [4,5,6]. AMPARs are (predominantly) postsynaptic proteins involved in the generation of excitatory currents and thus vital for physiological functions within the CNS, such as neurotransmission and synaptic plasticity [7,8]. Aabs against the AMPAR GluR3 subunit were originally discovered in patients with Rasmussen’s encephalitis (RE), a rare, predominantly paediatric, neurological disease associated with inflammation and seizures [9]. It has been further estimated that GluR3 Aabs are present in up to 20–30% of epilepsy patients [10], as well as in 20–25% of patients with frontotemporal dementia (FTD) [11]. It is typically considered that the presence of AMPAR Aabs in the cerebrospinal fluid is deleterious; however, a clear consensus on the processes triggering Aab production and the mechanisms by which they are implicated in neuronal excitability remains somewhat elusive [12].

Aabs generated by immunisation with GluR3-specific peptides have been studied in several animal models and reported to enhance seizures and be associated with behavioural abnormalities [9,13]. However, this was not fully reproducible in some other studies [14,15] and GluR3B Aabs were also shown to confer partial protection from seizures in rats [16]. Several in vitro mechanistic studies have led to different hypotheses as to how GluR3 Aabs mediate their effects. One prominent hypothesis is that Aabs bind to GluR3B-containing AMPARs and cause an agonist-like excitotoxic effect and subsequent neuronal death [15,17]. In line with these in vitro studies, histopathology showed an increase in neuronal death and associated reactive gliosis in response to GluR3B peptide immunisation in rats [16]. By contrast, in vitro studies in rat hippocampal neuronal primary cultures and human induced pluripotent stem cell (hiPSCs)-derived neurons from FTD patients have reported that anti-AMPAR Aabs caused a reduction in GluR3-containing AMPARs via increase in endocytosis and/or increased trafficking from cell surface, together with a reduction in dendritic spine density and, ultimately, decreased glutamate release [11,18,19]. Finally, others have reported no functional effect of patient anti-GluR3 Aabs on primary cortical neurons [20]. Therefore, a definitive answer as to whether the effects of anti-AMPAR Aabs are causal of, or an attempt to protect against, disease has yet to be determined (see also [18]).

Against this background, we generated affinity-purified anti-AMPAR Aabs following GluR3B peptide immunisation and tested the hypothesis that Aabs would have acute and/or longer-term functional effects on excitatory currents in hippocampal neurons. Our data demonstrate an inhibitory effect on excitatory currents and contribute to our further understanding of pathways affected by anti-AMPAR Aabs.

## 2. Results

### 2.1. Generation and Characterisation of AMPAR Aabs

A rabbit was immunised with a 24 amino acid GluR3B peptide. This sequence corresponds to an extracellular hinge region within the amino terminal domain (ATD) termed GluR3B [15,21], considered to be an immunogenic region [14]. Bleeds were initially analysed using ELISA (Appendix A). Binding of the terminal sera to the immunisation peptide was detected as low as 1:100,000, indicating successful generation of an immune response to the GluR3B peptide. Protein A-purified total IgG was subjected to quality check by Western blot (Appendix A) and analysed via SDS-PAGE and Coomassie staining (Appendix A). 7.6 mg/mL total IgG was purified, of which anywhere between 1–10% is predicted to be AMPAR-specific [22]. ELISA analysis using the total IgG material revealed a strong immunogenic response to the immunisation peptide, with a minimal response shown against an irrelevant peptide (Figure 1A). To characterize the binding of anti-AMPAR Abs to the native receptors we used Western blot analysis on whole rat brain tissue lysates. Blots were probed with anti-AMPAR Aabs, a commercial anti-AMPAR antibody (cAMPAR, Alomone AGC-010), IgG (negative) control or a secondary antibody only (negative) control (each 1:100) (Figure 1B). Anti-AMPAR Aabs detected a major band at just above 100 kDa (predicted GluR3 molecular weight is 101 kDa) (Figure 1B). The cAMPAR antibody detected a similar major band; a band < 75 kDa was also seen, the latter being consistent with reports of a GluR3 ‘short form’ detected by commercial antibodies [23]. Fainter bands at higher molecular weight may represent different glycosylated states of the protein [23]. The class-specific negative control rIgG did not show any clear bands at the predicted size for GluR3, rather, multiple bands at different molecular weights were detected; this may be due to the control being from a naïve non-immunised rabbit and likely containing other antibodies. No bands were detected in secondary-only controls. In order to link to functional experiments described below, primary neuronal cortical cultures were fixed and incubated with anti-AMPAR Aabs (Figure 1C) or cAMPAR or negative controls: rIgG and secondary antibody-only (each 1:100) (Appendix A). Anti-AMPAR Aabs gave clear staining of primary cortical neurons as indicated by co-labelling with the neuronal marker βIII tubulin (1:500) (Figure 1C), but not with the astrocyte marker GFAP (1:400). This staining was similar to that exhibited by cAMPAR, which also co-labelled with βIII tubulin, but not GFAP (Appendix A). No staining was detected for secondary antibody-only controls; class-specific negative control rIgG elicited only minor, faint staining (Appendix A). Together, these data demonstrate that anti-AMPAR Aabs exhibit target-specific binding to its native protein in neurons, similar to that observed with a commercial anti-AMPAR antibody.

### 2.2. Assessing the Functionality of Anti-AMPAR Aabs on Spontaneous Excitatory Postsynaptic Currents (sEPSCs)

As introduced above, anti-AMPAR Aabs have been reported to exert functional electrophysiological effects via targeting the GluR3 extracellular loop. AMPARs underlie fast excitatory neurotransmission in the CNS, where GluR3 subunits make an important contribution to EPSCs in hippocampal neurons [24]. Therefore, we determined functional effects of anti-AMPAR Aabs applied acutely (10 or 30 min) or longer-term (24 h) on sEPSCs in primary hippocampal neurons (DIV7-14, shown to express AMPARs, Figure 1) as an appropriate paradigm to investigate how these Aabs may affect neuronal activity. 

#### 2.2.1. Effects of Acute Anti-AMPAR Aab Incubation

Acute bath application (10 min) of anti-AMPAR Aabs (1:1000) caused a significant reduction in mean sEPSC frequency (0.8 ± 0.4 Hz) compared to those incubated with rIgG (1:1000) (1.3 ± 1.0 Hz; *n* = 21–23/group, *p* = 0.0396 unpaired *t*-test, Figure 2A). A corresponding significant difference in cumulative inter-event interval distributions was also observed for anti-AMPAR Aabs compared to rIgG 10 min incubation (*p* < 0.0001; Kolmogorov–Smirnov test; Figure 2B). Selected sEPSC traces of both rIgG- and anti-AMPAR Aab-treated cells are shown in Figure 2C. Here, and in other experiments, events were confirmed as AMPAR-mediated on the basis of sensitivity to the AMPAR antagonist, NBQX. No significant effect on mean sEPSC amplitude was observed following 10 min application of anti-AMPAR Aabs, whereby cells incubated with anti-AMPAR Aabs had sEPSCs of similar mean amplitude (−22.3 ± 7.3 pA) to those incubated with rIgG (−20.4 ± 7.2 pA, *n* = 24/group, unpaired *t*-test, *p* = 0.364; Figure 2D). Here, and in other experiments, there were no clear changes in event parameters including time to peak, half-width, rise time and decay kinetics (see Figure 2C and Appendix A).

To assess if effects on sEPSC frequency could be observed following a longer, but still ‘acute’, Aab incubation, anti-AMPAR Aabs or rIgG (1:1000) were bath applied for 30 min. AMPAR Aabs caused a significant reduction in mean sEPSC frequency (0.7 ± 0.4 Hz) compared to those cells incubated with rIgG (1.3 ± 0.8 Hz, *n* = 15/group, *p* = 0.0318; unpaired *t*-test, Figure 3A). A corresponding significant difference in cumulative inter-event interval distributions was also seen for anti-AMPAR Aabs compared to rIgG 30 min incubation (*p* < 0.0001; Kolmogorov–Smirnov test; Figure 3B). No significant effect was seen on mean sEPSC amplitude following application of anti-AMPAR Aabs 30 min incubation (*n* = 15/group, *p* = 0.889, unpaired *t*-test; Figure 3C). 

#### 2.2.2. Effects of 24 h Anti-AMPAR Aab Incubation

Exposure of native AMPARs to anti-AMPAR Aabs in patients is typically more ‘chronic’ in nature due to their presence in CSF [10]. Therefore, AMPAR Aabs or rIgG (1:1000) were applied to hippocampal neurons in culture 24 h prior to recording to investigate longer-term effects on sEPSCs.

Anti-AMPAR Aab-incubated cells showed a significant reduction in mean sEPSC frequency (0.4 ± 0.3 Hz) compared to rIgG incubated cells (1.2 ± 0.9 Hz, *n* = 11–13/group, *p* = 0.0113, unpaired *t*-test; Figure 4A). A corresponding significant difference in cumulative inter-event interval distributions was also observed following 24 h incubation with anti-AMPAR Aabs compared to control IgG-incubated cells (*p* < 0.0001; Kolmogorov–Smirnov test; Figure 4B). No difference in mean sEPSC amplitude was observed; cells incubated for 24 h with anti-AMPAR Aabs had sEPSCs of similar amplitude (−27.1 ± 8.1 pA) to those cells incubated with rIgG −21.8 ± 8.9 pA, *n* = 12/group, *p* = 0.142, unpaired *t*-test; Figure 4C).

### 2.3. Effects of Anti-AMPAR Aab Incubation on Miniature Excitatory Postsynaptic Currents (mEPSCs)

To explore the potential effects of anti-AMPAR Aabs on GluR3-mediated quantal transmitter release in hippocampal neurons (see [24]), we next determined any functional effects when applied acutely (30 min) or longer-term (24 h) on action potential-independent mEPSCS.

#### 2.3.1. Effects of Acute Anti-AMPAR Aab Incubation

Acute bath application (30 min) of AMPAR Aabs (1:1000) caused a significant reduction in mean mEPSC frequency (0.3 ± 0.2 Hz) compared to those incubated with rIgG (1:1000) (1.1 ± 1.0 Hz, *n* = 11/group, *p* = 0.0109, unpaired *t*-test; Figure 5A). A corresponding significant difference in cumulative mEPSC inter-event interval distributions was also observed for anti-AMPAR Aabs vs. rIgG 30 min incubation (*p* < 0.0001; Kolmogorov–Smirnov test; Figure 5B). There was no significant difference in mEPSC amplitude between anti-AMPAR Aabs vs. rIgG (30 min; unpaired *t*-test, *p* = 0.27, *n* = 12–14 per group).

#### 2.3.2. Effects of 24 h Anti-AMPAR Aab Incubation

Longer-term incubation (24 h) of AMPAR Aabs (1:1000) also caused a significant reduction in mean mEPSC frequency (0.4 ± 0.2 Hz) compared to those incubated with rIgG (1:1000) for 24 h (0.9 ± 0.6 Hz, *n* = 8/group, *p* = 0.0377, unpaired *t*-test; Figure 5C). A corresponding significant difference in cumulative mEPSC inter-event interval distributions was also observed for anti-AMPAR Aabs vs. rIgG 24 h incubation (*p* < 0.0001; Kolmogorov–Smirnov test; Figure 5D). Selected mEPSC traces of both rIgG and anti-AMPAR Aab treated cells are shown in Figure 5E. There was no significant difference in mEPSC amplitude between anti-AMPAR Aabs vs. rIgG (24 h; unpaired *t*-test, *p* = 0.87, *n* = 14–17 per group).

Overall, these data demonstrate a significant reduction in sEPSC and mEPSC frequency following both acute (10 and 30 min) and 24 h incubation of primary hippocampal neurons to anti-AMPAR Aab compared to control IgG. No changes in s/mEPSC amplitude were seen in any of the above conditions. Taken together, these data are consistent with an inhibitory anti-AMPAR Aab effect on excitatory currents.

## 3. Discussion

Aabs directed against subunits of AMPARs have been increasingly identified in the CNS of patients with a range of neuropathophysiologies, including encephalitis, dementia and epilepsy; however, Aab mechanisms of action are still not fully understood. This study has demonstrated the successful generation of anti-AMPAR Aabs following peptide immunisation, which successfully labelled native AMPARs and which had functional inhibitory effects on excitatory currents in hippocampal neurons.

### 3.1. Anti-AMPAR Aabs Bind to Native AMPARs

Specific Aabs directed against the AMPAR GluR3 subunit were successfully generated following immunisation with a peptide of a specific 24 amino acid sequence ATD-epitope within the GluR3B subunit. Protein A-purified Aabs identified native AMPARs, as shown via ELISA, immunocytochemistry and Western blot. In particular, the specificity of AMPAR Aabs for targets in hippocampal neurons was demonstrated by the co-localisation of AMPAR Aabs with βIII tubulin-labelled, but not GFAP-labelled cells, indicating a neuron-specific labelling for these Aabs. These results are similar to those reported previously, whereby peptide-derived anti-AMPAR Aabs successfully immunolabelled primary neurons [15]; moreover, recombinantly expressed GluR3 subunits were shown to be labelled by patient-derived anti-GluR3 Aabs [11]. Together, these data suggest that the anti-AMPAR Aabs produced here represent a valid experimental tool.

### 3.2. Anti-AMPAR Aabs Exhibit a Functional Inhibitory Effect

Our data demonstrates that acute or longer-term application of anti-AMPAR Aabs caused a consistent reduction in s/mEPSC frequency, with no effect on s/mEPSC amplitude, in hippocampal neurons. These data may suggest a presynaptic locus of action for anti-AMPAR Aabs, either via reduction in glutamate release or an alteration in the density of synaptic vesicles [25] or, alternatively, via Aab-induced receptor internalization (Figure 6). It is of note that we report similar functional effects in response to either acute (10–30 min) or longer-term (24 h) Aab exposure. In this regard, Haselmann et al. (2018) [26] also report that 24 h incubation with anti-GluR2 Aabs caused a reduction in mEPSC frequency, but not amplitude; similar effects were reported following ~1 h Aab application, but not for shorter (~30 min) applications [26]. Haselmann et al. (2018) further demonstrated that these effects were due to Aab-mediated receptor endocytosis and that internalization of AMPARs led to deficits in synaptic transmission [26].

Our data are most consistent with recent studies investigating the functional effects of GluR3 Aabs present in CSF from FTD patients [18]. Most notably, 30 min incubation with anti-GluR3 Aabs was shown to cause a dose-dependent decrease in AMPAR-evoked glutamate exocytosis in synaptosomes from mouse hippocampus [18]. Anti-GluR3 Aabs were also shown to decrease postsynaptic localisation of GluR3 subunits and cause a loss of dendritic spines, both in hippocampal neurons and in differentiated neurons from hiPSCs following 24 h exposure [11]. Further mechanistic studies reported that an increase in endocytosis of GluR3-containing AMPARs was accompanied by an increase in protein interacting with C kinase-1:glutamate receptor-interacting protein-1 ratio (proteins necessary for AMPAR internalization and insertion, respectively) [6,18]. Anti-AMPAR Aabs have been shown to promote an increase in endocytosis of GluR3-containing AMPARs in prefrontal cortex neurons [19], with similar effects on receptor internalization also reported for GluR1/2 Aabs [26,27] and NMDA NR1 Aabs [28]. Intracerebroventricular infusion of anti-GluR3 IgG purified from the serum of FTD patients was shown to reduce synaptic levels of GluR3-containing AMPARs in the prefrontal cortex, but not in the hippocampus in mice [19]. At the human level, FTD patients with cerebrospinal fluid anti-GluA3 antibodies also had reduced AMPAR levels in fractions purified from the temporal cortex [18]. Such reports are consistent with anti-AMPAR Aabs having functional inhibitory effects.

An alternative putative explanation of our data is that inhibition in transmitter release may reflect a reduction in the number of functional presynaptic neurons following anti-AMPAR Aab application (Figure 6). Earlier studies with recombinant anti-AMPAR Aabs and/or those from patient CSF, have proposed that acute Aab application caused an agonist-like effect leading to neuronal cell death. Thus, rabbit GluR3 antisera and IgG evoked CNQX-sensitive currents in a subset of cortical neurons [17]; co-application of a GluR3B peptide blocked this activity. A similar agonist effect was seen when anti-GluR3 Aabs were applied to rat neocortical brain slices; whole-cell currents were blocked by CNQX, but not by DL-APV, suggesting these Aabs act specifically on AMPARs [15]. Acute application of AMPAR Aabs also generated a rapid inward current in GluR3-expressing oocytes [29]. It is possible that these alternative effects reflect differences in dose, affinity and/or specificity to the anti-AMPAR Aabs generated in our study; furthermore, our Aabs are affinity-purified, any contamination with non-AMPAR Aabs in other studies may produce alternative effects; for example, AMPAR can be dynamically modulated through allosteric changes [30].

Longer-term (24 h) application of GluR3B Aabs in primary hippocampal neurons resulted in a significant increase in cell death [15]. In line with these in vitro studies, histopathology showed an increase in neuronal death and associated reactive gliosis in response to longer term GluR3B peptide immunisation in rats [16]. Thus, it is possible that treatment of hippocampal neurons with anti-AMPAR Aabs may result in cell death at presynaptic loci, which may also lead to the reduction in transmitter release reported here; however, the fact that we report similar effects under both acute (10–30 min) and longer term (24 h) exposure means that any cell death effects would need to occur over short time frames. 

Overall, we propose an inhibitory anti-AMPAR Aab effects which occur via a predominantly presynaptic action to reduce glutamate release and/or receptor internalization (Figure 6). The presence of presynaptically expressed GluR3 subunits in the hippocampus is now well documented, as is their potential contribution to pathophysiology [31]; thus, Zanetti et al. review several reports whereby GluR3, assembled with GluR2, make important contribution to presynaptic function, including transmitter release and receptor trafficking [31]. It is well known that AMPARs play an important role in synaptic plasticity and, in particular, expression on postsynaptic dendritic spines is a dynamic, activity-dependent process, which is regulated by distinct subunit composition and post-translational modifications [8]. Thus, it is possible that any reduction in postsynaptic GluR3 subunits contribution to AMPAR responses in our hippocampal neurons is poorly reflected here. However, it is widely reported that hippocampal neurons predominantly express either GluR1/2 or GluR2/3 heterodimers, with GluR2/3 making a smaller contribution to synaptic currents [24,32]. Moreover, GluR3 receptors subject to Aab-mediated internalization may be replaced by GluR1/GluR2-containing AMPARs in hippocampal neurons [33]. Overall, the in vivo situation is likely to be more complicated. In this regard, it has been shown that anti-AMPAR Aab-mediated decreases in excitatory currents were accompanied by a corresponding ‘homeostatic’ decrease in inhibitory currents and an overall increased intrinsic excitability [27]. Equally, combined actions of anti-AMPAR Aabs at pre- and postsynaptic GluR3 AMPARs are likely to result in a disruption to the excitatory/inhibitory network (as summarized in Figure 6).

### 3.3. Clinical Relevance

Studies in animal models have generally correlated the presence and/or introduction of anti-AMPAR Aabs with adverse behavioural changes, such as increased seizure-like activity [9,13]. An initial case study with one RE patient further reported that repeated plasma exchange to reduce the titre of anti-AMPAR Aabs resulted in an amelioration of RE symptoms [9]. However, there are also some conflicting results, with rat anti-AMPAR Aabs being shown to offer partial protection from PTZ-induced seizures in rats [16]. More recently, the presence of anti-AMPAR Aabs has been correlated with impairments in social behaviour and in social cognitive function in mice [19], the latter was proposed to be associated with neurodegenerative effects. The presence of GluR3B Aabs in epilepsy patients has also been correlated with neurological/psychiatric/behavioural abnormalities [34].

Overall, the anti-AMPAR Aab-induced inhibitory changes described here mirror the findings of other studies which demonstrate compromised AMPAR function. The molecular mechanisms by which anti-AMPAR Aabs act remain to be clarified fully, as does how such deficits in glutamatergic transmission manifest in RE, FTD and/or associated seizures. One straightforward explanation of these findings is that anti-AMPAR Aab-mediated reductions in glutamatergic transmission will dampen hyperexcitability. For example, transcranial magnetic stimulation in FTD patients revealed changes consistent with deficits in glutamatergic neurotransmission [18]; such studies support that a mechanistic reduction in excitatory currents may underlie disease phenotype. Going forward, developing more targeted treatments which normalise deficits in AMPAR function seen in the presence of anti-AMPAR Aab is likely to be beneficial in pathophysiological states.

## 4. Materials and Methods

### 4.1. Rabbit Immunisation and Antibody Production and Purification

An immunising peptide based on the human GluR3 amino acid 400–423 sequence (NEYERFVPFSDQQISNDSASSENR) was used to generate Aabs. The GluR3B peptide was modified with N-terminal acetylation and C-terminal amidation to help prevent degradation by exopeptidases. Peptides were conjugated to three different carrier proteins: keyhole limpet hemocyanin (KLH), bovine serum albumin (BSA) and ovalbumin (OVA) (Peptide Synthetics, Fareham, UK). A female New Zealand White rabbit (>2 kg) was immunized sub-cutaneously with the AMPAR GluR3B peptides described above (500 µg of peptide per immunization). For the initial priming immunisation, KLH-conjugated peptide was emulsified in an equal volume of Complete Freund’s Adjuvant (CFA). To drive a peptide-specific immune response, alternating booster injections were given at 14-day intervals using Incomplete Freund’s Adjuvant (IFA) with OVA- and then BSA-conjugated peptides, respectively. To monitor the serum response, bleeds were taken from the ear before the initial immunisation and before each boost and analysed as below. The rabbit was sacrificed 14 days after the final immunisation by Schedule 1 methods in accordance with the UK Animals (Scientific Procedures) Act 1986, at which time the spleen, bone marrow, peripheral blood mononuclear cells and lymph nodes were taken, along with the terminal serum.

#### 4.1.1. Serum Screening and Antibody Titre

ELISAs were carried out after each immunisation boost. 96-well microtiter plates were coated with streptavidin (2 µg/mL; Jackson ImmunoResearch, Cambridge, UK) overnight at 4 °C. Plates were washed (3×) with 1% PBS-Tween20 (PBS-T), blocked with 1% casein (VWR, Poole, UK) in 1% PBS-T (block buffer) for 1 h at room temperature, washed 3× 1% PBS-T and biotin-tagged peptides (1 µM) (ThermoFisher Scientific, Loughborough, UK) in block buffer were added to the wells and incubated at room temperature for a further 1 h. Bleed 0, 1, 2, 3 (pre-immunisation, post 1st, 2nd and 3rd boost) and terminal serum were added to the wells as a half-log dilution series, incubated for 1 h and subsequently washed (3×) with 1% PBS-T. Peptide-sera complexes were detected using an HRP-conjugated secondary antibody (1:4000, 45 min; Jackson ImmunoResearch, Cambridge, UK) for 1 h at room temperature. Complexes were washed 3× 1% PBS-T, incubated with 3,3′,5,5′ tetramethylbenzidine (Sigma Aldrich, Gillingham, UK) at room temperature and the reaction stopped by the addition of 2.5% NaF (Sigma Aldrich, Gillingham, UK). Absorbance was measured at 630 nm using a microplate reader (Synergy 5; BioTek, Swindon, UK).

#### 4.1.2. Protein A Purification of Polyclonal IgG Antibody from Rabbit Serum

Immunised rabbit terminal serum was purified using protein A resin (GE Healthcare, Chalfont St. Giles, Buckinghamshire, UK). Protein A-Sepharose beads (Sigma Aldrich, Gillingham, UK) were added to a 20 mL column and washed with 5× 10 mL PBS. Debris in the terminal serum was removed by filtration prior to addition to the column. The resin was re-suspended and mixed with the serum and left to mix gently on a roller overnight at 4 °C. The serum and resin were re-added to the column and the flow-through collected. The resin was washed 2× 50 mL PBS and antibodies eluted with 0.1 M sodium citrate (pH 3.5). 12× 8 mL fractions were collected in 1.2 mL 2 M Tris-HCl (pH 8.4) for pH neutralization (Sigma Aldrich, Gillingham, UK). Fractions containing antibodies were then pooled, buffer exchanged with PBS and concentrated using 10 kDa molecular weight cut off filters (Amicon; Sigma Aldrich, Gillingham, UK). Total IgG concentration was determined by absorbance at 280 nm using an A280 nanodrop (ThermoFisher Scientific, Loughborough, UK).

#### 4.1.3. SDS-PAGE Analysis of Aab Purity

Samples were combined with loading buffer (2× sample buffer), added to each well of a 4–20% Novex Tris-Glycine gel, and run alongside a pre-stained marker (all reagents, ThermoFisher Scientific, Loughborough, UK). The gel was post-stained with Coomassie Blue (Generon, Slough, UK) for 1 h, de-stained overnight using ddH_2_O and visualised using an ImageQuant LAS 4000 mini system (GE Healthcare, Chalfont St. Giles, Buckinghamshire, UK). 

### 4.2. Immunocytochemistry

Anti-AMPAR Aab specificity was assessed using immunocytochemistry. Primary and secondary antibodies used were as follows: rabbit anti-AMPAR (1:100; raised against residues 60–73 of rat GluR3 ATD, AGC-010, Alomone Labs, Jerusalem, Israel); rabbit anti-IgG_1_ (rIgG, 1:100, 011-000-003, Jackson ImmunoResearch, Cambridge, UK); mouse anti-IgG_2b_ (1:100, 70–4732, BioLegend, London, UK); mouse anti-βIII-tubulin (mIgG2b, 1:500, 801201, BioLegend, London, UK); mouse anti-glial fibrillary acidic protein (GFAP) (1:400, MAB3402, Millipore); goat anti-rabbit or anti-mouse Alexa Fluor 488/594/647 (all at 1:1000, Life Technologies, Loughborough, UK). For primary neurons, cells were washed (3×) with PBS and fixed with PFA for 10 min and washed 3× with PBS. Primary antibodies (anti-AMPAR Aabs, rIgG or mIgG2b) were added (in blocking buffer; PBS; 10% normal goat serum) and incubated for 2 h at room temperature. Coverslips were washed and permeabilised using 0.1% Triton X-100 and subsequently rewashed. Additional primary antibodies were added (anti-βIII tubulin or GFAP) for 2 h at room temperature. The cells were subjected to further washes and subsequently specific Alexa Fluor-coupled secondary antibodies were added (in blocking buffer) and incubated for 30 min at room temperature. Following incubation, cells were washed 3× with PBS. Nuclei were counterstained using DAPI (1:10,000) and coverslips were mounted in ProLong anti-fade mounting medium (Molecular Probes, Loughborough, UK). Cells were visualised with an AxioImager A1 microscope) and Axiovision 4.6.3 imaging software (Zeiss, Cambridge, UK).

#### SDS PAGE and Western Blotting

For preparation of protein lysates from whole brain, mouse (C57BL6/J, male 4–6 weeks) frozen brain hemispheres were defrosted in 2 mL ice-cold lysis buffer (150 mM NaCl, 1% (*v*/*v*) Triton-X-100, 10% (*v*/*v*) glycerol; 30 mM HEPES; SigmaFAST protease inhibitor 1 tablet/50 mL) and homogenised using an upright homogeniser (Lysing Matrix D, MP Biomedical, Santa Ana, CA, USA). Lysates were centrifuged at 14,000 rpm at 4 °C for 10 min and the supernatant removed, aliquoted and stored at −20 °C if not being used immediately. A bicinchoninic acid protein assay kit (ThermoFisher Scientific, Loughborough, UK) was used to determine the concentration of protein lysates. SDS-PAGE gels were prepared with 10% separating gel and 3% stacking gels. Samples were made up to the appropriate concentration using a loading buffer/2-mercaptoethanol mix and denatured at 90 °C for 5 min. Samples and molecular weight markers (Precision Plus pre-stained marker; Bio-Rad, Watford, UK) were loaded and SDS-PAGE gel electrophoresis was performed in running buffer (25 mM Tris base, 190 mM glycine, 0.1% *w*/*v* SDS) at a constant current of 0.02A. Polyvinylidene fluoride (PVDF, BioRad, Watford, UK) membranes for protein transfer were first activated by immersion in 100% methanol for ~15 s, transferred to ddH_2_O for 2 min and subsequently equilibrated in transfer buffer (25 mM Tris base, 190 mM glycine, 20% *v*/*v* methanol) for 10 min on a rocker. Transfer was carried out at a constant current of 0.2 A for 2 h. Membranes were fixed in methanol for ~15 s, followed by 5 min in 1× PBS on a rocker. PBS was replaced with blocking buffer (PBS; 5% non-fat milk powder, 1% Tween 20) and incubated for 1 h at room temperature on a rocker. Membranes were incubated overnight on a rocker at 4 °C with appropriate concentrations of primary antibody diluted in blocking buffer. The following day, membranes were washed with 1× PBS-T for 5 min (6×) on a rocker and incubated with HRP-conjugated secondary antibody for 1 h at room temperature (goat anti-mouse HRP-conjugated or goat anti-rabbit HRP-conjugated; 1:10,000, SeraCare). Finally, membranes were washed 6× 5 min with PBS-T before incubation in enhanced chemiluminescence detection buffer (Pierce; ThermoFisher Scientific, Loughborough, UK) for at least 5 min in the dark. Membranes were imaged using an ImageQuant LAS 4000 mini system (GE Healthcare, Chalfont St. Giles, Buckinghamshire, UK). All reagents, Sigma Aldrich, Gillingham, UK, unless stated.

### 4.3. Animals

The housing and use of animals in all experiments were carried out in accordance with UK Home Office regulations under the Animals (Scientific Procedures) Act, 1986. Male and female C57BL/6 mice (Charles River Ltd., London, UK) were used throughout all experiments at embryonic day 18. Mice were housed at 21 °C in a 12 h light/dark cycle with food and water available ad libitum. Experiments followed ARRIVE guidelines [35]. 

#### E18 Primary Neuronal Cell Culture

Embryonic day 18 (E18) C57BL/6 mice were used for primary neuronal cultures. Embryos were removed from the abdominal cavity of the adult female mouse following cervical dislocation. Heads were removed and placed in dissection media (DMEM-F12; Sigma Aldrich, Gillingham, UK) and cortex or hippocampi were chemically dissociated using papain (20 min incubation at 37 °C) and DNase (both Sigma Aldrich, Gillingham, UK) (2 mg/mL in PBS) for 30 s. Cells were transferred to pre-warmed culture medium (Neurobasal medium, 1% B27, 2 mM GlutaMax, 2.5% FBS, 100 U/mL/100 µg/mL penicillin/streptomycin; all Gibco/Life Technologies, Loughborough, UK) and gently triturated using needles (3 × 21 G followed by 3 × 23 G). Both cell suspensions were topped up to 5 mL and centrifuged at 250 rcf for 5 min to pellet the cells. The pellet resuspended to achieve a seeding density of 1.5 × 10^5^ cells/well or 2 × 10^5^ cells/well (hippocampus and cortex, respectively) of a 24-well plate on laminin-coated coverslips (Sigma Aldrich, Gillingham, UK). Cells were subjected to a 50% medium change after 2–3 days in culture.

### 4.4. In Vitro Electrophysiology 

The functionality of anti-AMPAR Aabs (1:1000) was further investigated using whole-cell patch-clamp on primary hippocampal neurons from day in vitro (DIV) 7–14. Patch pipettes were fabricated from borosilicate glass using a P1000 Flaming/Brown micropipette puller (WPI, Hitchin, Hertfordshire, UK) and had resistance of ~3–10 MΩ when filled with an intracellular solution consisting of: 145 mM K gluconate; 5 mM NaCl; 10 mM HEPES; 0.3 mM NaGTP; 4 mM MgATP; pH to 7.3, osmolarity 280 mOsm. sEPSCs were recorded from a holding potential of −70 mV in an extracellular solution consisting of: 130 mM NaCl; 3 mM KCl; 10 mM HEPES; 1 mM MgCl_2_; 2 mM CaCl_2_; 30 mM glucose, pH 7.3 and in the presence of GABA_A_R antagonist bicuculline methiodide (BMI; 10 µM) and NMDAR antagonist DL-2-amino-5-phosphonopentanoic acid (DL-APV; 50 µM). mEPSCs were recorded from a holding potential of −70 mV in the additional presence of tetrodotoxin (TTX; 1 μM, Abcam, Cambridge, UK). EPSCs were acquired using a MultiClamp 700B (Molecular Devices, Wokingham, UK) with Clampex 10.6 software (Molecular Devices, Wokingham, UK). Series resistance and membrane capacitance were recorded and monitored throughout the experiment; any significant changes in these parameters resulted in the recording being discarded. Current signals were filtered at 2 kHz and sampled at 10 kHz using an Axon Digidata 1550B. EPSCs were analysed using a template search function (Clampfit, Molecular Devices, Wokingham, UK).

For acute application experiments (10 min or 30 min), following a 10 min baseline, Aabs were added directly to the bath extracellular solution and incubated for 10 min or 30 min and EPSCs were monitored throughout. For longer-term application experiments (24 h), primary neurons were incubated with Aabs for 24 h prior to recording. For 24 h experiments, cells were placed in external solution with BMI and DL-APV as well as the respective antibody and allowed to equilibrate and for mEPSC experiments in the additional presence of TTX. For acute and long-term experiments, a negative control (rIgG) was used. At the end of recordings, the AMPAR antagonist 2,3-dioxo-6-nitro-7-sulfamoyl-benzo[f]quinoxaline (NBQX, 10 µM) was typically added to the bath to confirm that all observed events were mediated by AMPARs. All chemicals used here were from Sigma Alrich, Gillingham, UK unless stated.

### 4.5. Statistical Analysis

All data are presented as mean ± standard deviation (SD) with number of independent experiments (n) detailed in text and analysed using GraphPad Prism 7.00 (GraphPad Software, Inc., San Diego, CA, USA). Grubb’s outlier tests were performed to ensure that any outliers in the data set were removed prior to statistical analysis. All data sets were tested for normality using D’Agostino Pearson tests; on this basis, parametric paired and unpaired *t*-tests were performed to compare EPSC amplitude and frequency for anti-AMPAR Aabs to those incubated with the negative control rIgG; for further biophysical parameters including time to peak, half-width, rise time and decay time, multiple data sets were analysed using one-way ANOVA. Cumulative frequency plots were generated, whereby inter-event intervals (20 ms bins) were compared between Aab-incubated cells and control. These plots were analysed statistically by performing Kolmogorov–Smirnov tests. Throughout, data were considered significant at *p* < 0.05.

## Figures and Tables

**Figure 1 pharmaceuticals-16-00077-f001:**
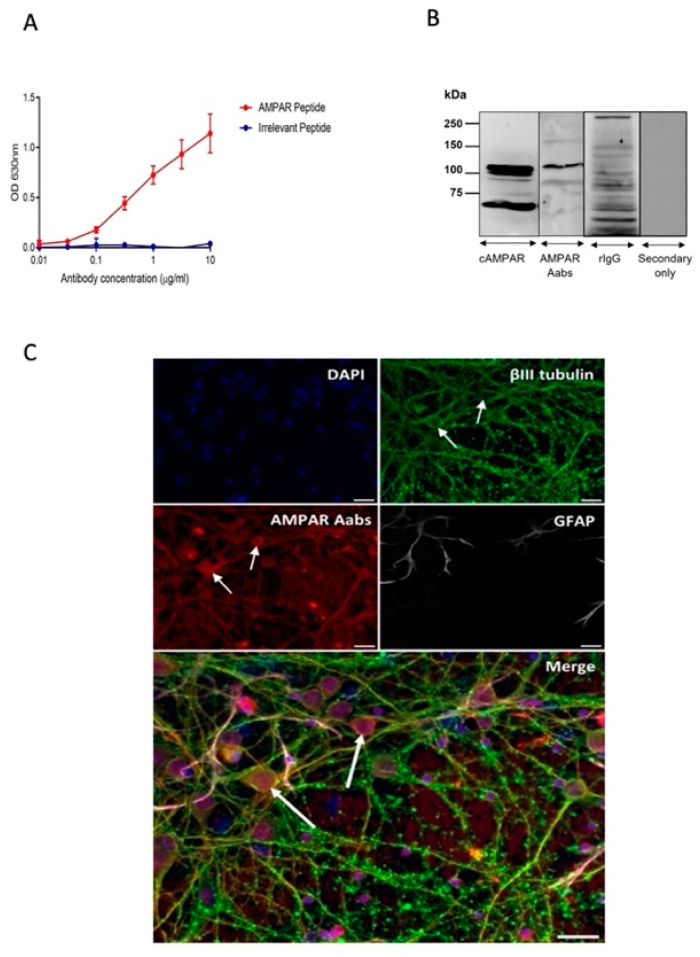
Rabbit anti-AMPAR GluR3B immunogenicity response. (**A**) ELISA of protein A-purified AMPAR (GluR3) Aabs against the GluR3 immunisation peptide or an irrelevant peptide; *n* = 3 technical replicates. (**B**) Western blot of mouse whole brain lysate probed with a commercial anti-AMPAR antibody (cAMPAR), anti-AMPAR Aabs, a class-specific negative control rIgG (naïve) or secondary antibody only (negative control). Representative blots from *n* = 3 technical replicates. (**C**) Immunocytochemical staining of fixed primary cortical neuron on cultures. Cells (DIV8) were stained with anti-AMPAR Aabs (red), anti-βIII tubulin (green), GFAP (white) and nuclei counterstained with DAPI (blue). Examples of labelling of hippocampal neurons with anti-AMPAR Aabs (red) is indicated by the white arrows. Scale bars = 20 μm. Representative images from *n* = 3 technical replicates.

**Figure 2 pharmaceuticals-16-00077-f002:**
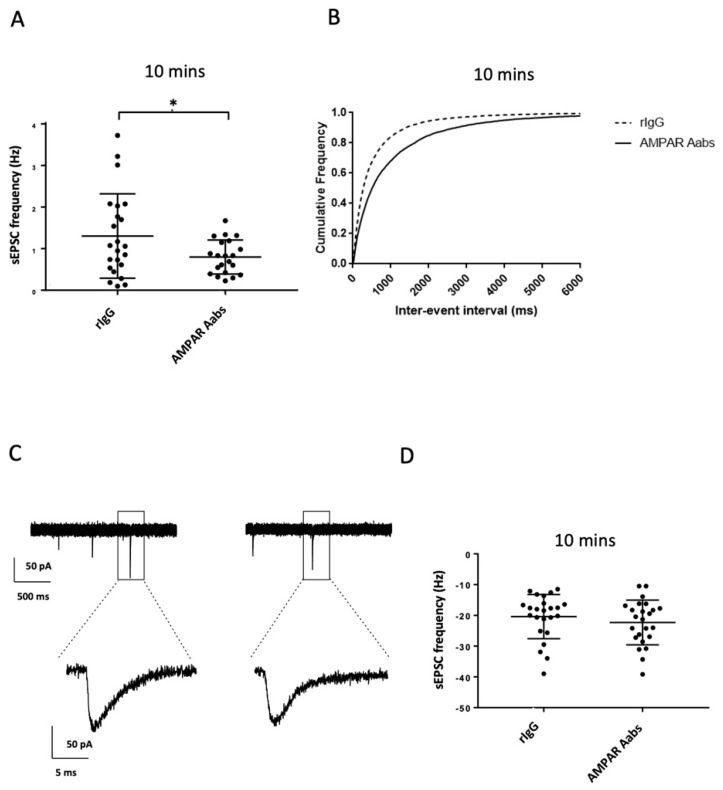
Effects of acute (10 min) anti-AMPAR Aabs and rIgG application on sEPSC frequency and amplitude. (**A**) Anti-AMPAR Aabs had a significantly lower mean sEPSC frequency than rIgG treated cells over the 10 min period (*p* < 0.05). (**B**) Significant differences in cumulative inter-event interval frequency were also observed for anti-AMPAR Aabs 10 min incubation compared to rIgG incubated cells (*p* < 0.0001). (**C**) Selected raw sEPSC traces for rIgG- and AMPAR-treated cells. (**D**) Anti-AMPAR Aabs had no significant effect on mean sEPSC amplitude vs rIgG treated cells over the 10 min period. Data were collected over three separate neuronal cultures, presented as mean ± SD and analysed by unpaired *t*-tests (**A**,**D**), or via Kolmogorov–Smirnov test (**C**). * = *p* < 0.05.

**Figure 3 pharmaceuticals-16-00077-f003:**
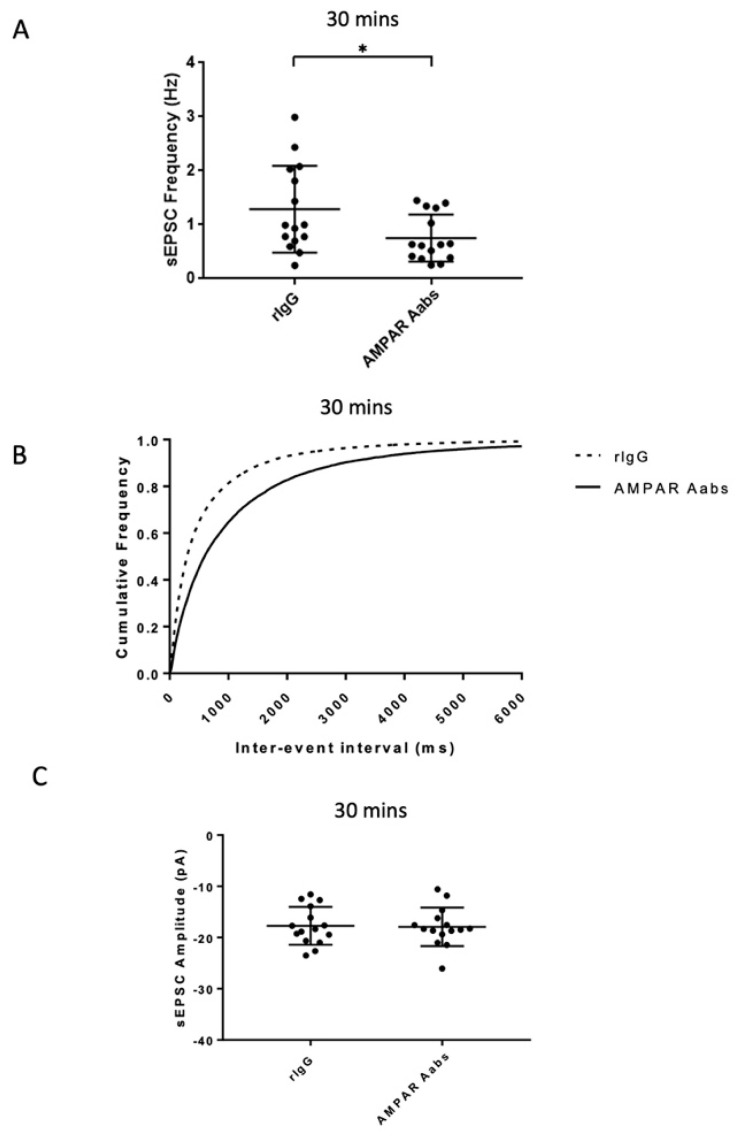
Effects of acute (30 min) anti-AMPAR Aabs and rIgG application on sEPSC frequency and amplitude. (**A**) Anti-AMPAR Aabs had a significantly lower mean sEPSC frequency than rIgG treated cells over the 30 min period (*p* < 0.05). (**B**) Significant differences in cumulative inter-event interval frequency were also observed for anti-AMPAR Aabs 30 min incubation compared to rIgG incubated cells (*p* < 0.0001). (**C**) Anti-AMPAR Aabs had no significant effect on mean sEPSC amplitude vs. rIgG treated cells over the 30 min period. Data were collected over three separate neuronal cultures, presented as mean ± SD and analysed by unpaired *t*-tests (**A**,**C**), or via Kolmogorov–Smirnov test (**B**). * = *p* < 0.05.

**Figure 4 pharmaceuticals-16-00077-f004:**
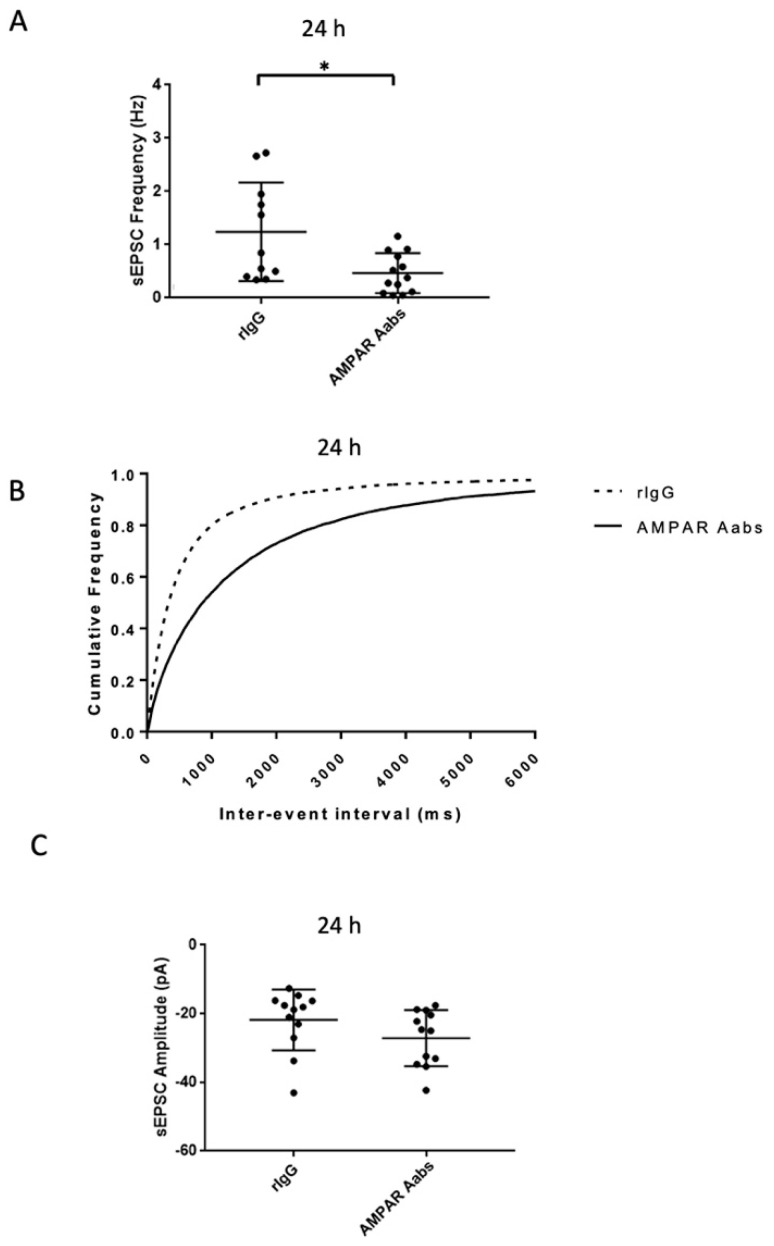
Effects of 24 h anti-AMPAR Aabs and rIgG application on sEPSC frequency and amplitude. (**A**) Anti-AMPAR Aabs had a significantly lower mean sEPSC frequency than rIgG treated cells over the 24 h period (*p* < 0.05). (**B**) Significant differences in cumulative inter-event interval frequency were also observed for anti-AMPAR Aabs 24 h incubation compared to rIgG incubated cells (*p* < 0.0001). (**C**) Anti-AMPAR Aabs had no significant effect on mean sEPSC amplitude vs. rIgG treated cells following 24 h incubation. Data were collected over three separate neuronal cultures, presented as mean ± SD and analysed by unpaired *t*-tests (**A**,**C**), or via Kolmogorov–Smirnov test (**B**). * = *p* < 0.05.

**Figure 5 pharmaceuticals-16-00077-f005:**
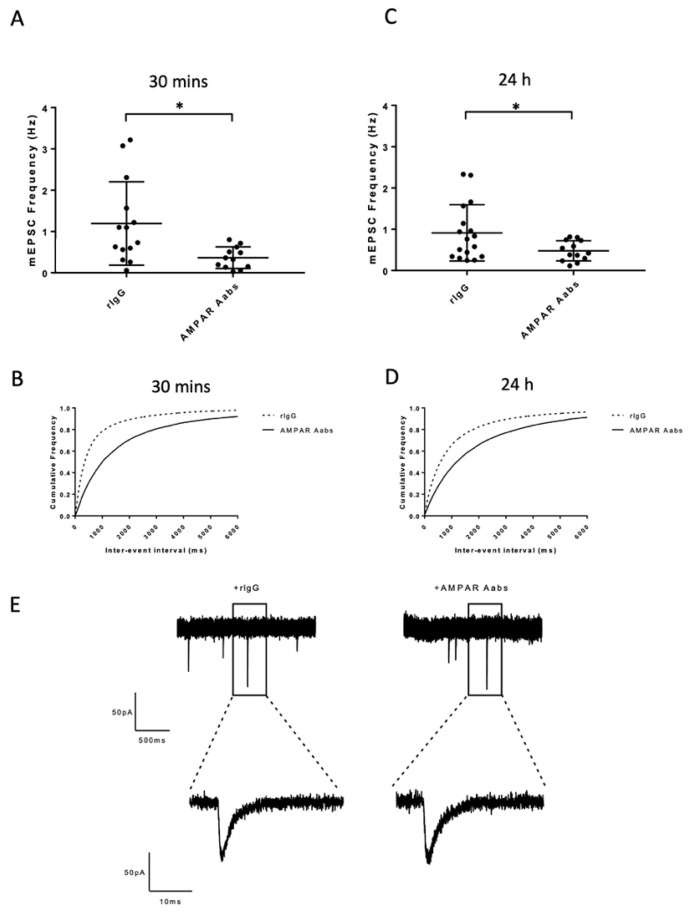
Effects of acute (30 min) and 24 h anti-AMPAR Aabs and rIgG application on mEPSC frequency and amplitude. (**A**) Anti-AMPAR Aabs had a significantly lower mean mEPSC frequency than rIgG treated cells over the 30 min period (*p* < 0.05). (**B**) Anti-AMPAR Aabs had a significantly lower mean mEPSC frequency than rIgG treated cells following 24 h incubation (*p* < 0.05). Significant differences in cumulative inter-event interval frequency were also observed for anti-AMPAR Aabs compared to rIgG incubated cells for (**C**) 30 min incubation and (**D**) following 24 h incubation (both *p* < 0.0001). Data were collected over three separate neuronal cultures, presented as mean ± SD and analysed by unpaired *t*-tests (**A**,**C**), or via Kolmogorov–Smirnov test (**B**,**D**). (**E**) Selected raw sEPSC traces for rIgG- and AMPAR-treated cells. * = *p* < 0.05.

**Figure 6 pharmaceuticals-16-00077-f006:**
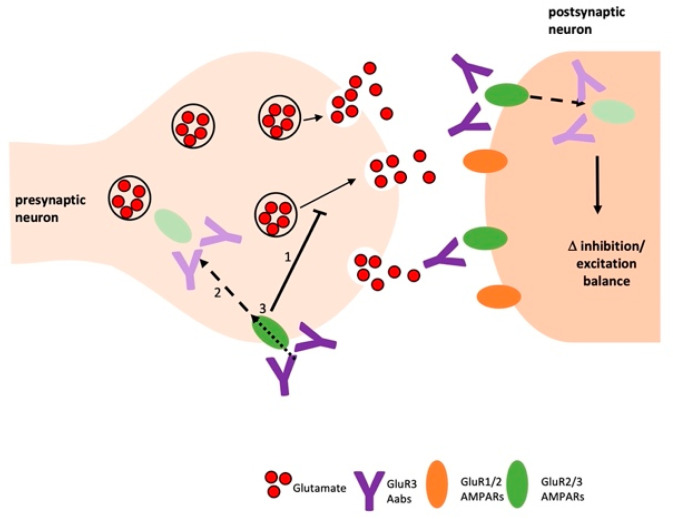
Schematic of potential mechanisms of anti-AMPAR Aabs. Anti-AMPAR Aabs may exert their inhibitory effect on glutamate release via presynaptic AMPARs. Additionally, anti-AMPAR Aabs may cause antibody-induced internalisation of AMPARs pre- and/or post-synaptically, resulting in an imbalance of the excitatory/inhibitory network.

## Data Availability

Data is contained within the article and Appendix A.

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
