# Peer review of "Anti-AMPA Receptor Autoantibodies Reduce Excitatory Currents in Rat Hippocampal Neurons"

_pharmaceuticals, 2023, doi:10.3390/ph16010077_

Round 1

Reviewer 1 Report

Overall, this is an interesting research article in which Day et al. have generated novel anti-AMPARs Aabs targeting peptides linked to autoimmune diseases demonstrating inhibitory effects on excitatory currents in hippocampal neurons. 

However, I have several comments:

1.       The structure of the results is confusing. One possibility is to structure like this:

2.1. Generation and characterisation of AMPAR Aabs

2.2. Assessing the functionality of anti-AMPAR Aabs on spontaneous excitatory postsynaptic currents  (sEPSCs).

2.2.1.  Effects of acute anti-AMPAR Aab incubation

2.2.2 Effects of 24 h anti-AMPAR Aab incubation

2.3. Effects of anti-AMPAR Aab incubation on miniature excitatory postsynaptic currents 191 (mEPSCs)

2.3.1.  Effects of acute anti-AMPAR Aab incubation

2.3.2 Effects of 24 h anti-AMPAR Aab incubation. Currently these results are incorrectly included in the " Effects of acute anti-AMPAR Aab incubation " subsection.

Another option would be to delete subsections 2.2.1, 2.2.2, 2.3.1 and 2.3.2.

2.       In results, line 136-138, you state that “Here and in other experiments, there were no clear changes in event activation and decay kinetics (see Figure 2C and Figure 5E). Did you measure the rise time, half-width, or the decay of the events? If you didn´t is too hazardous to state this sentence. It could be interesting to analyze these parameters to know if there are or there are not changes in events kinetics and duration. This additional information would reinforce the quality of the work.

3.       Other minor comments:

4.       What is the membrane holding stablished to study synaptic events? Please include it in the text.

5.       In figure 2A the axle titles are difficult to see.

6.       In figure 3 and 4 it could be better if you included the raw data traces of EPSC as in the other figures.

7.       Please change GLUR3B to GLuR3 (line 14)

8.       Abbreviations in text and figure legends. In general, you should review that you use the word or phrase in full when you refer the term for the first time followed by the abbreviation in parentheses and thereafter use the abbreviation only. Some examples are:

a.       CNS appears for the first time in line 32.

b.       Frontotemporal dementia is already abbreviated as FTD in line 40 (remove word in full in line 55)

c.       the meaning of ATD in line 71 is missing as well as the meaning of CSF in line 170.

d.       AMPA receptors should be AMPAR in line 260…..

Author Response

Overall, this is an interesting research article in which Day et al. have generated novel anti-AMPARs Aabs targeting peptides linked to autoimmune diseases demonstrating inhibitory effects on excitatory currents in hippocampal neurons. 

We thank the reviewer for positive comments and for useful suggestions; we address these points below and revise text (including a new Supplemental Figure 3) accordingly.

The structure of the results is confusing

We thank the reviewer for this useful suggestion. We have now correctly formatted the Results section, including differentiating effects on mEPSCs (we have similarly formatted the Discussion and Methods) and agree that this improves the presentation.

In results, line 136-138, you state that “Here and in other experiments, there were no clear changes in event activation and decay kinetics (see Figure 2C and Figure 5E). Did you measure the rise time, half-width, or the decay of the events? If you didn´t is too hazardous to state this sentence. It could be interesting to analyze these parameters to know if there are or there are not changes in events kinetics and duration. This additional information would reinforce the quality of the work.

We have measured these parameters for sEPSPs and, to address the Reviewer point, we now include box and whisker plots for each of time to peak, half-width, rise time and decay time as a new Supplemental Figure 3. These plots illustrate that baseline parameters of time to peak, half-width, rise time and decay time were unaffected by anti-AMPA Aabs and rIgG, and that there was no difference in any parameter between anti-AMPA Aabs and rIgG (analysed using one-way ANOVA, detail added to Methods) and we agree that these data supplement our findings.

Other minor comments:

What is the membrane holding stablished to study synaptic events? Please include it in the text.

Holding potential was -70 mV for sEPSCs and mEPSCs, this information has been added to the Methods

In figure 2A the axle titles are difficult to see.

Corrected, here and at other figures

In figure 3 and 4 it could be better if you included the raw data traces of EPSC as in the other figures.

We appreciated this point, but feel that further raw traces (which essential replicate those in Figure 2 and, to a lesser extent, Figure 5) do not add much to these figures. However, to address this point, we now include further raw traces as figures in the new Supplemental Figure 3; these illustrate overlay sEPSC events at baseline and in the presence of anti-AMPA Aabs (1:1000) and rIgG (1:1000) as fitted with a template search function. We hope that this addresses the reviewer concern.

Please change GLUR3B to GLuR3 (line 14)

The GluR3 subunit of AMPARs has been identified as a target for autoantibodies (Aabs) and, here, these Aabs were raised against a sequence which corresponds to an extracellular hinge region within the ATD termed GluR3B; thus, we raised Aabs using a GluR3B peptide. Moreover, different labs have used different terminology for these regions and the nomenclature can be confusing. For this specific point, Line 14 does not appear to contain “GluR3B” (we are unsure if the formatting on our copy is different from the reviewer version); however, we have now checked the paper thoroughly to make sure our terminology is consistent with either that used in the papers referred to, or to our own study, and believe that this is now corrected throughout.

Abbreviations in text and figure legends. In general, you should review that you use the word or phrase in full when you refer the term for the first time followed by the abbreviation in parentheses and thereafter use the abbreviation only. Some examples are:

CNS appears for the first time in line 32. Corrected

Frontotemporal dementia is already abbreviated as FTD in line 40 (remove word in full in line 55) Corrected

the meaning of ATD in line 71 is missing as well as the meaning of CSF in line 170. Corrected

AMPA receptors should be AMPAR in line 260….. Corrected

Thank you for these points, we have now checked the paper thoroughly and made these corrections.

Reviewer 2 Report

dear authors, thank you very much for the paper. 

i am confused as title said rat and methods said rabbit. can you please check this. this is not simple issue rat as a research model can be a rational solution between large animal models and typical laboratory mice because of their size, genetic homogenity, availability of genetically modified stains and possibility to perform research mimicking clinical applications. so kindly confirm. 

i noticed that Anti-cytokine autoantibodies (AAbs) not been abbreviated first use. 

in introduction i think authors need to explain better that both in healthy persons and in patients with a variety of clinical diseases, AAbs are frequent and involve a sizable panel of cytokines. Anti-cytokine AAbs are considered to be a normal component of the AAb repertoire in healthy persons and are assumed to help with the delicate control of cytokine homeostasis.

i did not follow the rationale very well i think authors need to clarify that the reasoning for this study. Excitatory glutamatergic synapses in the mammalian central nervous system use ion flow through alpha-amino-3-hydroxy-5-methyl-4-isoxazolepropionic acid receptors to transmit neurotransmission (AMPARs). AMPARs, which are extremely dynamic and move into and out of synapses in a manner depending on activity, are abundant in the postsynaptic membrane on dendritic spines. A variety of factors can control AMPARs, alter synaptic strength, and support cellular forms of learning. These factors include changes in their quantity, subunit composition, phosphorylation status, and auxiliary proteins. Furthermore, disruption of AMPAR plasticity has significant negative effects on mental health and has been linked to a number of psychiatric conditions. In this article, we concentrate on the mechanisms that regulate AMPAR plasticity, notably drawing on significant research on hippocampal synapses, and we highlight recent developments in the field while also outlining potential future paths. this should be part of the discussion too.

section 3.3. Clinical relevance is nicely written but need expansion. Presynaptic AMPARs appear to have a significant modulatory function in nerve terminal activity in the mammalian central nervous system, making them appealing as potential pharmaceutical targets for a number of clinical disorders.

in methods clarify how many was sample. 

Author Response

dear authors, thank you very much for the paper. 

We thank the reviewer for positive comments and for useful suggestions; we address these points below and revise text (including a new Supplemental Figure 3) accordingly.

i am confused as title said rat and methods said rabbit. can you please check this. this is not simple issue rat as a research model can be a rational solution between large animal models and typical laboratory mice because of their size, genetic homogenity, availability of genetically modified stains and possibility to perform research mimicking clinical applications. so kindly confirm.

To explain this point, we should point out that the anti-AMPAR Aabs are raised in rabbits using an immunising peptide based on the human GluR3 amino acid 400-423 sequence; the methodology of raising such antibodies in the rabbit species is usual practice. The electrophysiological investigation of these anti-AMPAR Aabs is in rat hippocampal neurons which, again, is a standard mammalian cell model to measure such effects. We hope that this clarifies this point.

i noticed that Anti-cytokine autoantibodies (AAbs) not been abbreviated first use. 

in introduction i think authors need to explain better that both in healthy persons and in patients with a variety of clinical diseases, AAbs are frequent and involve a sizable panel of cytokines. Anti-cytokine AAbs are considered to be a normal component of the AAb repertoire in healthy persons and are assumed to help with the delicate control of cytokine homeostasis.

We should also point out here that, although anti-cytokine autoantibodies are an important sub-set of antibodies and have been implicated in autoimmune responses, that they are not the focus of this study. Here we use ‘Aabs’ to define ‘autoantibodies’ as a more general term (on first use, Abstract line 14) and focus here on the clinical diseases attributed in the literature to involve specifically anti-AMPAR Aabs. We hope that this clarifies this point.

i did not follow the rationale very well i think authors need to clarify that the reasoning for this study. Excitatory glutamatergic synapses in the mammalian central nervous system use ion flow through alpha-amino-3-hydroxy-5-methyl-4-isoxazolepropionic acid receptors to transmit neurotransmission (AMPARs). AMPARs, which are extremely dynamic and move into and out of synapses in a manner depending on activity, are abundant in the postsynaptic membrane on dendritic spines. A variety of factors can control AMPARs, alter synaptic strength, and support cellular forms of learning. These factors include changes in their quantity, subunit composition, phosphorylation status, and auxiliary proteins. Furthermore, disruption of AMPAR plasticity has significant negative effects on mental health and has been linked to a number of psychiatric conditions. In this article, we concentrate on the mechanisms that regulate AMPAR plasticity, notably drawing on significant research on hippocampal synapses, and we highlight recent developments in the field while also outlining potential future paths. this should be part of the discussion too.

We thank the reviewer for this suggestion. In our Abstract, we outline that AMPARs are (predominantly) postsynaptic proteins involved in the generation of excitatory currents and thus vital for physiological functions within the CNS, such as neurotransmission and synaptic plasticity [7,8] (line 37-39). We should however point out that the GluR subtypes that are best associated with synaptic plasticity are the GluR1 (also termed GluA1) and GluR2 (also termed GluA2) subtypes, and not the GluR3 subtype studied here. However, we appreciate that our Discussion around control of insertion of different AMPAR subtypes at hippocampal neurons and, in particular, roles in determining synaptic strength and plasticity (original lines 326-333) can be strengthened. To address the reviewer point, we make additional reference to the review of Chater & Goda, 2014 and relate our findings to potential consequences on synaptic plasticity; thus, new lines 329-343:

It is well known that AMPARs play an important role in synaptic plasticity and, in particular, expression on postsynaptic dendritic spines is a dynamic, activity-dependent process, which is regulated by distinct subunit composition and post-translational modifications [8]. Thus, it is possible that any reduction in postsynaptic GluR3 subunits contribution to AMPAR responses in our hippocampal neurons is poorly reflected here. However, it is widely reported that hippocampal neurons predominantly express either GluR1/2 or GluR2/3 heterodimers, with GluR2/3 making a smaller contribution to synaptic currents [24,32]. Moreover, GluR3 receptors subject to Aab-mediated internalization may be replaced by GluR1/GluR2-containing AMPARs in hippocampal neurons [33].

section 3.3. Clinical relevance is nicely written but need expansion. Presynaptic AMPARs appear to have a significant modulatory function in nerve terminal activity in the mammalian central nervous system, making them appealing as potential pharmaceutical targets for a number of clinical disorders.

The reviewer makes a good point here. One strong possibility arising from our data is an effect at presynaptic AMPARs. We currently refer to a review by Zanetti et al which highlights potential contributions to pathophysiology [ref 31]. To address the reviewer point, we now extend this Discussion; thus, new lines 326-329:

Zanetti et al. review several reports whereby GluR3, assembled with GluR2, make important contribution to presynaptic function, including transmitter release and receptor trafficking [31].

in methods clarify how many was sample. 

The number of independent replicants (n) is given in the main text and illustrated via use of vertical scatter plots in electrophysiology figures. To address the reviewer point, we have added a sentence to indicate this in the Methods (line 524)

Round 2

Reviewer 2 Report

thank you for professionally addressing all concerns/queries.